# Exploring Contemporary Data on Lipid-Lowering Therapy Prescribing in Patients Following Discharge for Atherosclerotic Cardiovascular Disease in the South of Italy

**DOI:** 10.3390/jcm11154344

**Published:** 2022-07-26

**Authors:** Anna Citarella, Simona Cammarota, Francesca Futura Bernardi, Luigi Caliendo, Antonello D’Andrea, Biagio Fimiani, Marianna Fogliasecca, Daniela Pacella, Rita Pagnotta, Ugo Trama, Giovanni Battista Zito, Mariarosaria Cillo, Adriano Vercellone

**Affiliations:** 1LinkHealth Health Economics, Outcomes & Epidemiology S.R.L., 80143 Naples, Italy; simona.cammarota@linkhealth.it (S.C.); marianna.fogliasecca@linkhealth.it (M.F.); 2Regional Pharmaceutical Unit, Campania Region, 80143 Naples, Italy; bernardi.francesca.futura@gmail.com (F.F.B.); ugo.trama@regione.campania.it (U.T.); 3Department of Cardiology, Santa Maria della Pietà Hospital, Nola, 80035 Naples, Italy; l.caliendo@aslnapoli3sud.it; 4Department of Cardiology and Intensive Care Unit, Umberto I Hospital, Nocera Inferiore, 84014 Salerno, Italy; antonellodandrea@libero.it; 5Cardiology Service, Local Health Unit (LHU) Salerno, Associazioni Regionali Cardiologi Ambulatoriali (ARCA), 84129 Salerno, Italy; biagio.fimiani@tiscali.it; 6Department of Public Health, University of Naples Federico II, 80131 Naples, Italy; daniela.pacella@unina.it; 7Department of Management Control, Local Health Unit (LHU) Naples 3 South, 80053 Naples, Italy; r.pagnocta@aslnapoli3sud.it; 8Cardiology Service, Local Health Unit (LHU) Naples 3 South, Associazioni Regionali Cardiologi Ambulatoriali (ARCA), 80045 Naples, Italy; presidente@arcacardio.eu; 9Pharmaceutical Department, Local Health Unit (LHU) Salerno, 84124 Salerno, Italy; m.cillo@aslsalerno.it; 10Pharmaceutical Department, Local Health Unit (LHU) Naples 3 South, 80053 Naples, Italy; adriano.vercellone@libero.it

**Keywords:** lipid-lowering therapy, atherosclerotic cardiovascular disease, real-world data, high-intensity lipid-lowering therapy

## Abstract

Current international guidelines strongly recommend the use of high-intensity lipid-lowering therapy (LLT) after hospitalization for atherosclerotic cardiovascular disease (ASCVD) events. With this study, our aim was to evaluate LLT prescribing in a large Italian cohort of patients after discharge for an ASCVD event, exploring factors associated with a lower likelihood of receiving any LLT and high-intensity LLT. Individuals aged 18 years and older discharged for an ASCVD event in 2019–2020 were identified using hospital discharge abstracts from two local health units of the Campania region. LLT treatment patterns were analyzed in the 6 months after the index event. Logistic regression models were developed for estimating patient predictors of any LLT prescription and to compare high-intensity and low-to-moderate-intensity LLT. Results: A total of 8705 subjects were identified. In the 6 months post-discharge, 56.7% of patients were prescribed LLT and, of those, 48.7% were high-intensity LLT. Female sex, older age, and stroke/TIA or PAD conditions were associated with a higher likelihood of not receiving high-intensity LLT. Similar predictors were found for LLT prescriptions. LLT utilization and the specific use of high-intensity LLT remain low in patients with ASCVD, suggesting a substantial unmet need among these patients in the contemporary real-world setting.

## 1. Introduction

Atherosclerotic cardiovascular disease (ASCVD) remains the leading cause of death and disability in Europe, despite consistent improvement in outcomes [1]. As Europe’s population continues to age, ASCVD incidence is set to increase dramatically. Therefore, the prevention and treatment of the development and progression of ASCVD is a major public health issue that must be properly addressed.

Undoubtedly, the control of lipid levels is one of the most effective strategies for ASCVD event prevention [2]. Low-density lipoprotein cholesterol (LDL-C) is causally related to ASCVD and lowering LDL-C can significantly reduce ASCVD risk. LDL-C reduction with statins has been shown to reduce the risk of cardiovascular (CV) events, especially in individuals with established ASCVD [3,4,5,6]. More recently, clinical trial studies have shown that the addition of either ezetimibe or anti-proprotein convertase subtilisin/kexin type 9 inhibitor (PCSK9i) to statin therapy provides a further reduction in ASCVD risk [7,8,9]. In light of this, the 2019 European Society of Cardiology/European Atherosclerosis Society guidelines recommend more strict lipid management targets for established ASCVD patients to reach LDL-C <1.4 mmol/L (<55 mg/dL) and LDL-C reduction of 50% from baseline levels. To achieve the above goals, the initiation of high-intensity statin therapy is recommended, with consideration of add-on non-statin therapy (ezetimibe or PCSK9 inhibitors) if treatment targets are not met [10]. Moreover, these guidelines for the first time suggested the possibility of introducing PCSK9 inhibitors for acute coronary syndrome (ACS) patients during hospitalization [10].

Despite all this, lipid-lowering therapy (LLT) may remain underutilized or sub-optimally utilized in clinical practice in Europe [11,12,13]. In Italy, previous studies based on real-world data have highlighted scarce use and inappropriate choice of LLT by healthcare providers in high-CV-risk patients, especially in those with ischemic stroke and Peripheral Arterial Disease (PAD) [14,15,16,17]. However, the application of current guidelines is poorly understood in the management of ASCVD patients.

Therefore, our aim is to provide contemporary data on the prescribing of LLT in a large Italian cohort of patients post-discharge for an ASCVD event. Additionally, we explore demographics and clinical characteristics associated with a lower likelihood of receiving any LLT and, in particular, high-intensity LLT. Finally, we investigate any change in the prescribing of LLT before and after ASCVD discharge.

## 2. Methods

### 2.1. Data Source and Study Population

This retrospective observational study was conducted using administrative databases from two local health units (LHUs) in the Campania region (southern Italy), with a population of approximately 2 million health-assisted subjects. A record-linkage analysis of the civil registry, outpatient pharmaceutical databases, hospital discharge records (HDRs), and disease-specific exemption database was performed, including data from 1 January 2011 to 31 December 2020. Briefly, the civil registry contained all demographic data (including sex, date of birth and death) for the patients in the analysis. Outpatient pharmaceutical databases include dispensing date and anatomical therapeutic chemical (ATC) codes of drugs prescribed by specialists and primary care physicians. HDRs collect information on patients’ hospital discharge date, discharge diagnosis, and up to 5 secondary diagnoses and procedures recorded according to the International Classification of Diseases, Ninth Revision, Clinical Modification (ICD-9-CM). The disease-specific exemption database reports the date of exemption from copayment to healthcare and the disease according to ICD9 codes. All data were linked through the unique individual identification code properly anonymized to respect the subject’s privacy. Informed consent was not required to use encrypted retrospective information for research purposes as it is impossible to collect for organizational reasons. This study was submitted to and approved by the local ethics committees of the participating LHUs (N. 124, 24 July 2020; N. 13, 25 February 2020). The research adhered to the tenets of the Declaration of Helsinki.

For the study aim, we identified all residents of the two LHUs, aged 18 and older, with a hospital admission for an ASCVD episode during the period between 1 January 2019 and 30 June 2020 (enrollment period) (Figure 1). The ASCVD events were identified according to the revascularization procedures performed on the patients in the hospital or the discharge diagnosis including one of the following conditions: acute coronary syndrome (ACS), stable angina, unstable angina, stroke or transient ischemic attack (TIA), and PAD. The revascularization procedures were ascertained using the procedure codes for percutaneous coronary intervention (PCI), coronary artery bypass graft (CABG), or other revascularization procedures in the HDRs. ICD9 or procedure codes used for identifying ASCVD conditions are reported in the Appendix A. The date of the first hospital discharge for an ASCVD episode during the enrollment period was designated as the index date. All patients were followed up from for 6 months after the index date or until death or the end of study period (31 December 2020), whichever occurred first (follow-up period).

### 2.2. Lipid-Lowering Therapy Patterns

The LLT treatment pattern for each patient was identified on the basis of the first drug prescription collected within the 6 months after the index date. LLTs included in the study were statins, ezetimibe, and PCSK9i. The prescribing of LLT was grouped by (a) overall LLT, (b) high-intensity LLT (high-intensity statin and PCSK9i), and (c) low-/moderate-intensity LLT (low or moderate statin intensity), all either alone or in combination with ezetimibe. Statin was classified as high, moderate, or low intensity in accordance with American College of Cardiology (ACC)/American Heart Association (AHA) guidelines, as EAS/ESC guidelines do not define intensity [18]. High-intensity statin included atorvastatin 40 and 80 mg and rosuvastatin 20 and 40 mg; moderate-intensity statin was defined as atorvastatin 10–20 mg, rosuvastatin 5–10 mg, simvastatin 20–40 mg, pravastatin 40–80 mg, lovastatin 40 mg, fluvastatin 40–80 mg, pitavastatin 2–4 mg; low-intensity statin included all other statin types and doses.

### 2.3. Covariates

For each patient, the following variables were assessed at the index date: age group (i.e., 18–54, 55–64, 65–74, and 75 years and older), sex, ASCVD condition, and comorbidities. Patients were classified into ASCVD subpopulations representing the following conditions: ACS, stable angina, unstable angina, stroke or TIA, and PAD. To reduce the risk of multicollinearity in the regression models, the revascularization procedures category includes patients with only revascularization procedures without other ASCVD conditions, while patients with both ACS and procedure revascularization were collectively referred to as ‘ACS with procedure revascularization’. This approach was also used for stable and unstable angina. This method of categorization into mutually exclusive groups helped facilitate a comparison of findings between ASCVD subpopulations in a manner that is not influenced by an overlap between them.

The comorbidities identified for each patient during the last 8 years before the index date were diabetes, hypertension, congestive heart failure, chronic ischemic heart disease, other cerebrovascular disease, other cardiac disease, chronic kidney disease, familial hypercholesterolemia, and liver disease. The algorithms used for identifying the comorbidities are reported in the Appendix A. Moreover, we assessed the presence of ASCVD episodes identified in primary or secondary diagnosis in all HDRs during the last 2 years before the index date.

Additionally, we evaluated if the patients received at least one prescription of any LLT within 6 months before the index date. The last LLT prescription filled during the look-back period (1 year pre-index date) was evaluated to identify the LLT type (high-, moderate-, low-intensity statin or PCSK9i).

### 2.4. Statistical Analysis

Demographics, clinical characteristics, and LLT prescriptions were summarized as relative frequencies (percentage). A matrix analysis was performed to assess change patterns for prescribing LLT pre-and post-index date. The analysis showed changes ‘from’ (vertical axis) and ‘to’ (horizontal axis) for LLT prescribing (yes or no) and LLT type (high-, moderate-, low-intensity statin or PCSK9i).

Single and multiple logistic regression models were developed for estimating patient predictors of both LLT prescribing and LLT type. The model for the prediction of LLT prescribing was used to compare the predictors of LLT prescribing versus no LLT prescribing (reference). The model for predicting the LLT type was built to compare high-intensity LLT (high-intensity statin or iPCSK9) versus low-to-moderate-intensity LLT (low or moderate statin intensity). Age group at the index date, sex, ASCVD index, comorbidities, and number of previous ASCVD events 2 years before the index date (categorized as dichotomous variables) were evaluated as potential confounders. Factors that resulted significant at the simple regression model were added to the multiple logistic regression model to estimate adjusted odds ratios (ORs). The results are presented as estimates of odds ratios (ORs) with 95% confidence intervals (CIs). For all analyses, a *p* < 0.05 is considered significant. All analyses were performed using SPSS software (version 23.0, SPSS Inc., Chicago, IL, USA) and R statistical software (version 4.0.3, R Foundation for Statistical Computing, Vienna, Austria).

## 3. Results

A total of 8705 patients who required hospitalization for ASCVD were identified after applying the inclusion criteria. Of that, 57.9% were male and 63.6% were older than 65 years. In the overall cohort, 30.2% of patients had stroke/TIA, 27.7% had ACS (with or without revascularization procedures), 7% had stable angina (with or without revascularization procedures), 4.9% had unstable angina (with or without revascularization procedures), 8.9% had PAD, and 21.3% underwent a revascularization procedure. The most common comorbid conditions were hypertension (62.5%), followed by diabetes mellitus (43.6%). The 18.7% of the cohort were hospitalized for an ASCVD event over 2 years before their index event. Demographic and clinical characteristics of the ASCVD patients at the discharge index are reported in Table 1.

In the 6-month period immediately after ASCVD event discharge, 56.7% of patients (4934 subjects) had filled a prescription for LLT (Figure 2). Stratifying by LLT type, 51.3% received low-/moderate-intensity LLT, of which 49.5% received moderate statin intensity (2441 subjects) and 1.8% low statin intensity (90 subjects), and 48.7% received high-intensity LLT, of which 47.4% received high statin intensity (2339 subjects) and 1.3% PCSK9i (64 subjects) (Figure 2). The Appendix A reports the demographic and clinical characteristics of ASCVD patients by LLT type.

Figure 3 shows the prescribing and the intensity of LLT (low/moderate or high intensity) stratified by age, sex, and ASCVD condition. The proportion of male patients who received at least one LLT prescription was higher than females in all age groups, with a major difference in the subjects aged between 18 and 54 years (Figure 3A). The proportion of patients with high-intensity LLT decreases with increasing age and was higher in males rather than females in all age groups (Figure 3B). Stratifying by ASCVD condition, the proportion of patients with LLT prescriptions and high-intensity LLT was over 70% for those who underwent a revascularization procedure and a primary diagnosis for unstable angina, stable angina, or ACS (Figure 3C). For both LLT prescriptions and high-intensity LLT, the smallest proportions were reported for subjects with a primary diagnosis of stroke/TIA (49.4% and 24.5%, respectively) or PAD (44.8% and 33.1%, respectively) (Figure 3C,D).

Table 2 shows the results of unadjusted and multivariable logistic models to evaluate the association between demographics and clinical factors and the prescribing of LLT versus no LLT, as well as high-intensity versus low-to-moderate-intensity LLT. Increasing age, being female, having familial hypercholesterolemia, and ASCVD event in the 2 years before index discharge increased the odds of receiving LLT prescriptions (Table 2). Among the ASCVD conditions, the strongest predictors for LLT utilization were ACS with or without revascularization procedures, unstable angina with and without revascularization procedures, and stable angina with revascularization procedures. Patients with a primary diagnosis for PAD had a lower likelihood of being treated with LLT (OR 0.70; 95%CI 0.59–0.84).

Similarly, the adjusted ORs of receiving high-intensity LLT were significantly higher in female patients (OR = 1.46; 95%CI: 1.27–1.67) discharged for ACS with (OR 4.66; 95%CI 3.74–5.80) or without (OR 3.52; 95%CI 2.86–4.34) revascularization procedures, unstable angina with (OR 5.35; 95%CI 3.05–9.37) or without (OR 2.48; 95%CI 1.79–3.44) procedures, and stable angina with procedures (OR 3.20; 95%CI 1.84–5.57). Conversely, patients with stroke/TIA or PAD had a lower likelihood of being treated with high-intensity LLT. Likewise, being ≥75 years old decreased by 40% the odds of being prescribed a high-intensity LLT.

Matrix analysis for LLT therapies during the 1-year look-back period and the 6-month post-discharge period is provided in Table 3. Among subjects not treated with LLT during the 6 months before their ASCVD hospitalization, 35.8% filled an LLT prescription post-discharge (40.3% low-/moderate-intensity LLT, 58.7% high-intensity statin, and 1.1% PCSK9i). Regarding the intensity of the LLT pre- and post-ASCVD episode, 72.1% confirmed low-/moderate-intensity and 27.9% changed to high-intensity LLT (27.0% high-intensity statin and 0.9% PCSK9i). On the other hand, among high-intensity LLT users pre-ASCVD discharge, 87.7% remained on the high-intensity LLT and 12.3% changed to low-/moderate-intensity LLT.

## 4. Discussion

This study provides contemporary real-world data on LLT utilization in patients following discharge for ASCVD in southern Italy. At the time of the study, the 2019 ESC guidelines strongly recommend LLT utilization in ASCVD patients and, particularly, the early initiation of high-intensity LLT to reduce CV risk in this population [10]. Recently, the 2021 ESC guideline include the same recommendations [19]. Despite that, in our study, we found that a large proportion of subjects (43%) did not receive any LLT following ASCVD discharge, and among LLT-treated patients, only 49% received a high-intensity LLT as recommended in the current treatment guidelines. The strongest predictors of filling a high-intensity LLT prescription after discharge for ASCVD event were the cardiac conditions (ACS, stable angina, unstable angina) with or without revascularization procedures. Conversely, the patients discharged for stroke/TIA and PAD had the lowest likelihood to be treated with the recommended LLT.

The relevance of our study is strengthened by the size of our sample and the ability to draw information from a real-world setting. Indeed, the use of administrative databases is a strength in real-world studies such as this one, and measuring drug exposure by filled prescriptions not only eliminated recall bias, but also improved accuracy of the information on drug use. Nevertheless, we acknowledge that several potential limitations might have influenced our results. As in administrative data sources, type and number of clinical variables are limited. Disadvantages are the lack of some clinical measurements useful for the definition of patients’ characteristics and prognostic stratification. A good deal of information on the degree of lipid control which may influence the likelihood of receiving high-intensity LLT is not available in our data sources. Therefore, we were not able to explore the use and type of LLT in relation to the achievement of the recommended goals of LDL-C. However, 2019 ESC guidelines have moved towards recommending at least a 50% reduction in LDL-C, in addition to achieving specific LDL-C goals for those at highest risk. This is intended to discourage the use of less intense statin regimens in patients at very high risk whose LDL-C levels are close to goal, and who, using a goal-based approach alone, could potentially receive suboptimal LDL-C reduction, despite apparently being at goal.

Despite these limitations, the data collected in our study are consistent with previous real-world European data highlighting unmet treatment needs in the contemporary era. Marz et al. found that only 44.9% of ASCVD patients received any LLT in Germany, and the proportion of patients with a current prescription for statins was highest among recent ACS patients (58.7%) and lowest among patients with PAD (34.8%) [20].

Sciattella et al. found that only 47% of subjects received a high-intensity LLT when analyzing healthcare claims data (2009–2014) for 17,881 Italian patients within 6 months after the first hospitalization for acute coronary syndrome, ischemic stroke, or PAD [21]. However, in that context, the proportion of patients without LLT was lower than ours (only 6% of patients were untreated with LLT). Furthermore, Arca et al. observed that only 39% of Italian ASCVD patients received any LLT in 2015, of whom only 12% had a high-intensity prescription, highlighting the low level of lipid goal attainment in clinical practice [22]. Additionally, in our study we found 90 subjects treated with low-intensity statin after discharge for ASCVD event (1.8% on the total of ASCVD patients treated with LLT). This finding is in line with the result from the EU-Wide Cross-Sectional Observational Study of Lipid-Modifying Therapy Use in Secondary and Primary Care (the DA VINCI study) that reported 2% of established ASCVD patients treated with low-intensity statin [23]. Interestingly, the DA VINCI study provides an opportunity to assess the impact of ESC guideline updates on clinical practice reflecting the difficulty of achieving more stringent goals. In that study, the LDL-C goal attainment in very high-risk patients fell by approximately half to only 11% after the changes in ESC recommendations between 2016 and 2019. Although in our study we cannot evaluate the rate of patients in whom the target LDL-C goals have been reached, the recent data also clearly show that, in most cases, only combination therapy with statins, ezetimibe, and PCSK9 inhibitors might be on target for most ASCVD patients [23,24]. In fact, the 2019 ESC guidelines suggest—especially in the specific context of ACS patients, and patients whose LDL-C levels are not at goal levels despite already taking a maximally tolerated statin dose and ezetimibe—adding a PCSK9i early after the event (if possible, during hospitalization for the ACS event) [10]. In our study, ezetimibe was used in 10.3% of patients, respectively, and PCSK9i used in combination with statins and/or ezetimibe in 1.3% of patients (1.5% of patients post-discharge for ACS or stable angina) (data not shown). The proportion of patients treated with PCSK9i increased to 2.9% in the subgroup of subjects with at least one ASCVD hospitalization during the pre-index period (data not shown).

Regarding ASCVD conditions, as reported in other real-world studies, we found that LLT utilization was lower among patients with stroke/TIA or PAD versus those with coronary heart disease (CHD). In particular, we showed that 50% and 45% of patients with stroke/TIA or PAD received an LLT prescription, respectively. Among these, only 25% and 33% of individuals with stroke/TIA or PVD, respectively, had a high-intensity LLT, suggesting that these remain neglected conditions in the contemporary era. Similarly, Steen et al. showed that patients with indications of ischemic stroke or PAD were less likely to receive statins than those with coronary conditions in the U.S. [12]. More recently, Colantonio et al. revealed that 33.9% and 43.0% with PAD only and cerebrovascular disease only, respectively, were taking a statin [25]. Although the authors found a similar risk for ASCVD event recurrence, a small proportion of those with PAD or cerebrovascular disease without CHD were taking a high-intensity statin compared to those with CHD. These results confirm a gap in care and a missed opportunity for ASCVD risk reduction in these populations and the need for more intensive secondary prevention interventions [26,27]. Furthermore, analyzing the information about hospital wards (available for 54% of the study population) we found that the proportion of patients with stroke/TIA who received high-intensity statin therapy was 20% and 60% when they were discharged by the neurology and cardiology ward, respectively (data not shown). These findings may suggest that the cardiologists, who presumably were more up-to-date on the efficacy of the LLT therapy, may have a major awareness of ESC guidelines.

Additionally, despite the demonstrated benefit of statins in diabetic patients, mainly with ASCVD, in our study the presence of diabetes is not associated with LLT use and type [19]. In our study, we found that only 57% of diabetic patients received LLT following ASCVD discharge, and among LLT-treated diabetic patients, only 45% received a high-intensity LLT as recommended in the current treatment guidelines.

While, in accordance with previous studies, we found that age and sex were important factors associated with LLT use. In particular, compared with patients aged <55 years, we found that older patients (≥75 years old) were more likely to receive any LLT, but were less likely to have a high-intensity LLT post-discharge. In particular, in our study we found that 53.3% of low-intensity statin users were older than 75 years (Appendix A). Despite the evidence that statins reduce major cardiovascular events irrespective of age, clinicians should be more judicious in providing high-intensity LLT to older patients, given the higher prevalence of multi-comorbidities, potential adverse drug effects, drug interactions due to polypharmacy, and limited life expectancy in the older population [28,29]. Furthermore, we found that women have approximately 27% lower odds than men to be treated with any LLT. Among those treated, women also had 50% lower odds of receiving a high-potency therapy even after adjusting for age and clinical conditions. We found that 63.3% of low-intensity statin users were female (Appendix A). Our findings are congruent with previous observations showing that women with CHD are less likely to receive state-of-the-art, evidence-based treatment than men [30,31,32,33,34]. Health care providers may perceive women with ASCVD to be at a lower risk of recurrent ASCVD event than their male counterparts. However, although the prevalence of ASCVD is higher in men, ASCVD mortality is actually higher among women, and there is no evidence that LLTs are less effective for secondary prevention or not tolerated as well by women as men [5,35,36,37,38,39,40,41]. Additionally, the fear of doctors and patients about possible side effects of drugs, reported more often by women, may influence decisions about using these drugs in lower doses [42,43,44].

Finally, consistent with other studies, our findings suggest that an important challenge remains: the intensification of lipid-lowering therapy following discharge. Indeed, we observed that 88% of patients treated with high-intensity LLT in the pre-hospital period confirmed the same LLT intensity after discharge, while the remaining 12% changed to low-/moderate-intensity LLT. On the other hand, only the 27% of patients on low-/moderate-intensity LLT before hospitalization changed to high-intensity LLT after discharge. However, administrative claims do not contain data that would permit a systematic evaluation of the characteristics that may have influenced the continuation of the pre-hospital statin regimen following discharge.

In conclusion, the LLT utilization and the specific use of high-intensity LLT remain low in patients post-discharge for ASCVD event, suggesting a substantial unmet need among these patients in the contemporary real-world setting. In the absence of LDL values, we cannot evaluate the rate of patients in whom the target LDL levels have been reached according to the guideline recommendations. However, the type of LLT observed in this study suggests that the recommendation to use high-intensity LLT was not adequately followed. Female sex, older age, and stroke/TIA or PAD conditions are associated with a higher likelihood of not receiving a high-intensity LLT. These predictors may be useful in planning to optimize the management of ASCVD patients. Many monitoring strategies on the patient and hospital levels could be implemented to enhance lipid-lowering treatment rates following discharge and appropriate management.

## Figures and Tables

**Figure 1 jcm-11-04344-f001:**
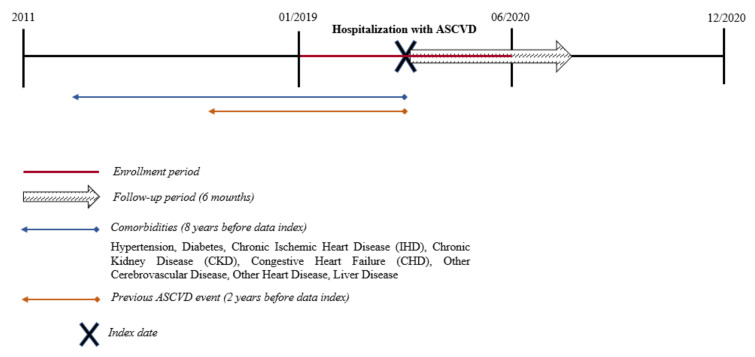
Study design. Atherosclerotic Cardiovascular Disease (ASCVD): Acute coronary syndrome (ACS), stable angina, unstable angina, stroke or transient ischemic attack (TIA), peripheral arterial disease (PAD), and revascularization procedures (percutaneous coronary intervention (PCI), coronary artery bypass graft (CABG) or other revascularization procedures).

**Figure 2 jcm-11-04344-f002:**
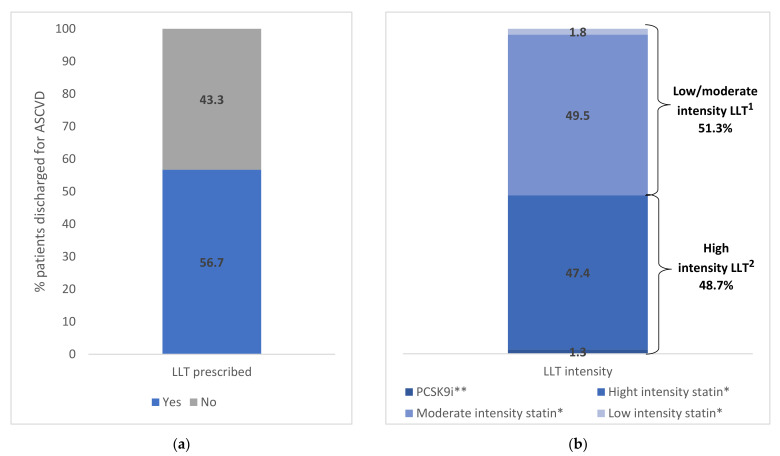
(**a**) Lipid Lowering Therapy (LLT) in ASCVD patients within 6 months from the discharge. (**b**) Lipid Lowering Therapy (LLT) in ASCVD patients within 6 months from the discharge stratified by LDL-C category. * With and without ezetimibe. ** With and without ezetimibe and/or statin therapy. ^1^ High-intensity LLT included atorvastatin 40–80 mg, rosuvastatin 20–40 mg and PCSK9i. ^2^ Low-/moderate-intensity statin included all other statin types and doses. Abbreviations: ASCVD, atherosclerotic cardiovascular disease; LLT, LDL-C-lowering therapy; PCSK9i, proprotein convertase subtilisin/kexin 9 inhibitor.

**Figure 3 jcm-11-04344-f003:**
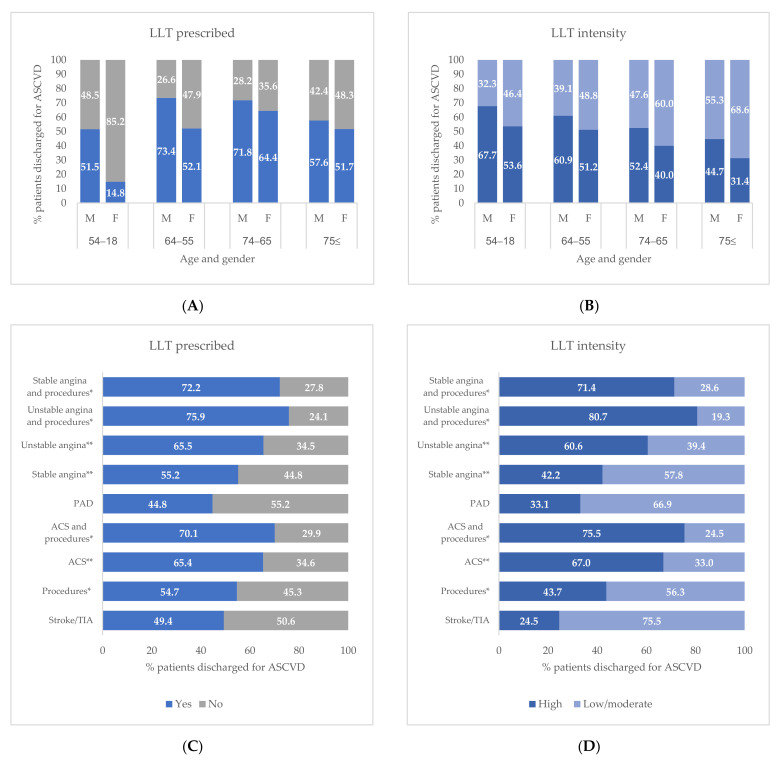
Lipid Lowering Therapy (LLT) in ASCVD patients within 6 months from the discharge stratified by sex and age (**A**) and ASCVD condition (**B**). Type of Lipid Lowering Therapy (LLT) in ASCVD patients within 6 months from the discharge stratified by sex and age (**C**) and ASCVD condition (**D**). * Procedures including percutaneous coronary intervention, coronary artery bypass surgery, or other revascularization procedures. ** Without procedures. Abbreviations: ACS, acute coronary syndrome; ASCVD, atherosclerotic cardiovascular disease; PAD, peripheral arterial disease; TIA, transient ischemic attack.

**Table 1 jcm-11-04344-t001:** Demographic and clinical characteristics for the study population.

	**ASCVD Patients**	
	**N = 8705**	
	**N**	**%**
**Gender**		
Male	5044	57.9
Female	3661	42.1
**Age (years)**		
18–54	1627	18.7
55–64	1541	17.7
65–74	2197	25.2
≥75	3340	38.4
**ASCVD condition**		
Stroke/TIA	2626	30.2
Procedures ^1^	1854	21.3
ACS ^2^	1256	14.4
ACS and procedures ^1^	1160	13.3
PAD	775	8.9
Stable angina ^2^	511	5.9
Unstable angina ^2^	310	3.6
Unstable angina and procedures ^1^	116	1.3
Stable angina and procedures ^1^	97	1.1
**Comorbidities**		
Hypertension	5438	62.5
Diabetes	3797	43.6
Chronic ischemic heart disease	3000	34.5
Congestive heart failure	1888	21.7
Chronic kidney disease	1791	20.6
Other cerebrovascular disease	1759	20.2
Other heart disease	1045	12.0
Familial hypercholesterolemia	100	1.0
Liver disease	89	1.0
**Previous ASCVD event**	1624	18.7

^1^ Procedures including percutaneous coronary intervention, coronary artery bypass surgery, or other revascularization procedures. ^2^ Without procedures. Abbreviations: ACS, acute coronary syndrome; ASCVD, atherosclerotic cardiovascular disease; PAD, peripheral arterial disease; TIA, transient ischemic attack.

**Table 2 jcm-11-04344-t002:** Demographic and clinical factors associated with LLT prescribing (ref. not prescribing) and with high-intensity LLT prescribing (ref. low-/moderate-intensity LLT prescribing).

	LLT Prescription		High-Intensity LLT Prescription	
Covariates	OR (95%CI)	Adjusted OR (95%CI)	OR (95%CI)	Adjusted OR (95%CI)
**Female gender (ref. male)**	2.02 (1.85–2.20)	1.73 (1.57–1.90)	1.92 (1.70–2.16)	1.46 (1.27–1.67)
**Age (ref. <55 years)**				
55–64	3.79 (3.27–4.39)	3.64 (3.11–4.27)	0.76 (0.62–0.94)	0.95 (0.75–1.20)
65–74	4.28 (3.74–4.91)	4.65 (4.01–5.39)	0.51 (0.42–0.62)	0.77 (0.62–0.97)
≥75	2.29 (2.03–2.59)	3.06 (2.67–3.51)	0.34 (0.28–0.41)	0.60 (0.48–0.76)
**ASCVD condition**				
Stroke/TIA	0.65 (0.60–0.72)	0.89 (0.79–1.02)	0.24 (0.21–0.28)	0.58 (0.48–0.71)
Procedures ^1^	0.90 (0.81–1.00)	NA ^3^	NA ^3^	NA ^3^
ACS ^2^	1.53 (1.35–1.74)	1.99 (1.70–2.33)	2.48 (2.11–2.90)	3.52 (2.86–4.34)
ACS and procedures ^1^	1.95 (1.70–2.23)	2.48 (2.11–2.93)	4.02 (3.39–4.77)	4.66 (3.74–5.80)
PAD	0.59 (0.51–0.69)	0.70 (0.59–0.84)	0.50 (0.40–0.63)	0.74 (0.56–0.97)
Stable angina ^2^	0.94 (0.78–1.12)	NA ^3^	0.76 (0.59–0.97)	1.00 (0.75–1.32)
Unstable angina ^2^	1.47 (1.16–1.87)	1.88 (1.44–2.47)	1.65 (1.24–2.20)	2.48 (1.79–3.44)
Unstable angina and procedures ^1^	2.43 (1.61–3.79)	2.54 (1.61–4.10)	4.50 (2.64–7.67)	5.35 (3.05–9.37)
Stable angina and procedures ^1^	2.00 (1.29–3.17)	1.83 (1.14–3.01)	2.67 (1.58–4.49)	3.20 (1.84–5.57)
**Comorbidities**				
Hypertension	0.89 (0.82–0.97)	0.93 (0.84–1.03)	0.67 (0.59–0.75)	0.72 (0.63–0.82)
Diabetes	1.06 (0.97–1.15)	NA ^3^	0.78 (0.70–0.88)	0.91 (0.80–1.03)
Chronic ischemic heart disease	1.55 (1.42–1.70)	1.36 (1.22–1.51)	1.68 (1.49–1.88)	1.62 (1.14–1.87)
Congestive heart failure	0.48 (0.43–0.53)	0.46 (0.41–0.52)	0.73 (0.63–0.85)	0.70 (0.58–0.84)
Chronic kidney disease	0.47 (0.42–0.52)	0.56 (0.50–0.64)	0.55 (0.47–0.65)	0.67 (0.55–0.81)
Other cerebrovascular disease	0.59 (0.53–0.66)	0.75 (0.66–0.84)	0.40 (0.34–0.47)	0.72 (0.60–0.87)
Other heart disease	0.74 (0.65–0.85)	0.84 (0.72–0.97)	0.77 (0.65–0.93)	0.79 (0.64–0.98)
Familial hypercholesterolemia	4.76 (2.70–8.39)	4.18 (2.40–7.83)	0.79 (0.51–1.22)	NA ^3^
Liver disease	0.94 (0.62–1.43)	NA ^3^	0.51 (0.28–0.92)	0.63 (0.33–1.22)
**Previous ASCVD event**	1.56 (1.40–1.75)	1.92 (1.68–2.20)	1.18 (1.03–1.36)	1.57 (1.33–1.86)

^1^ Procedures including percutaneous coronary intervention, coronary artery bypass surgery, or other revascularization procedures. ^2^ Without procedures. **^3^** The variable has been excluded from the multiple regression model due to its limited representation in this sample outcome subgroup that led to biased estimation for correlation-based multicollinearity. Abbreviations: ACS, acute coronary syndrome; ASCVD, atherosclerotic cardiovascular disease; CI, confidence intervals; OR, odds ratio; PAD, peripheral arterial disease; TIA, transient ischemic attack.

**Table 3 jcm-11-04344-t003:** Matrix analysis of LDL-C-lowering drug therapies.

Pre-ASCVD Discharge	Post-ASCVD Discharge					
	LLT		Intensity LLT ^1^			
	No	Yes	Low-Intensity Statin ^2^	Moderate-Intensity Statin ^2^	High-Intensity Statin ^2^	PCSK9i ^3^
	%	%	%	%	%	%
No LLT (N = 5043)	64.2	35.8	0.7	39.6	58.7	1.1
LLT (N = 3662)	14.6	85.4	2.5	55.2	40.9	1.4
Low-intensity statin * (N = 162)	20.4	79.6	55.0	22.5	21.7	0.8
Moderate-intensity statin * (N = 2672)	15.9	84.1	0.3	71.5	27.3	0.9
High-intensity statin * (N = 814)	9.2	90.8	0.1	12.2	86.3	1.4
PCSK9i ** (N = 14)	0.0	100.0	0.0	0.0	0.0	100.0

^1^ % calculated on group pre-ASCVD discharge. ^2^ With and without ezetimibe. ^3^ With and without ezetimibe and/or statin therapies. High-intensity statin: atorvastatin 40–80 mg, rosuvastatin 20–40 mg. Moderate-intensity statin: atorvastatin 10–20 mg, fluvastatin 80 mg, lovastatin 40 mg, pitavastatin 2–4 mg, pravastatin 40–80 mg, rosuvastatin 5–10 mg, simvastatin 20–40 mg. Low-intensity statin: fluvastatin 20–40 mg, lovastatin 20 mg, pitavastatin 1 mg, pravastatin 10–20 mg, simvastatin 10 mg. Abbreviations: ASCVD, atherosclerotic cardiovascular disease; LLT, LDL-C-lowering therapy; PCSK9i, proprotein convertase subtilisin kexin 9 inhibitors; * With and without ezetimibe; ** With and without ezetimibe and/or statin therapy.

## Data Availability

Not applicable.

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
