# Peer review of "Exploring Contemporary Data on Lipid-Lowering Therapy Prescribing in Patients Following Discharge for Atherosclerotic Cardiovascular Disease in the South of Italy"

_jcm, 2022, doi:10.3390/jcm11154344_

Round 1
Reviewer 1 Report
My comments have been adequately addressed. The manuscript has been improved significantly.
Reviewer 2 Report
In the light of the current 2021 ESC Guidelines on cardiovascular disease prevention in clinical practice, European Heart Journal, the most important thing is to achieve the therapeutic target of LDL cholesterol according to the cardiovascular risk class. The current article does not contain any data on LDL cholesterol, either during hospitalization or during the observation period. In my opinion, this is crucial when assessing hypolipemic treatment.
This manuscript is a resubmission of an earlier submission. The following is a list of the peer review reports and author responses from that submission.
Round 1
Reviewer 1 Report
The manuscript is very well written. The authors described based on real data the gap existing between real world practice and guidelines.
Please find below further comments regarding the manuscript, which could improve it:
1. How could the authors explain the low rate of high intensity statin therapy in patients with stroke and PAD? Perhaps, that cardiologists are more motivated with statins and these patients are hospitalized in non-cardiology clinics? Please comment
2. Did the authors include in their analysis patients with all types of stroke? e.g lacunar, cardioembolic, large artery atherosclerosis?
3. Is there any information regarding LDL levels pre-hospitalization or post hospitalization? Is there any information regarding the percentage of patients that have achieved LDL targets?
4. How many patients received PCSK9 therapy? All those patients were already under therapy with ezetimibe?
Author Response
How could the authors explain the low rate of high intensity statin therapy in patients with stroke and PAD? Perhaps, that cardiologists are more motivated with statins and these patients are hospitalized in non-cardiology clinics? Please comment
Thanks for your comment. The information about hospital wards was available only for 54% of study population. We analysed these data to explored the effect of the hospital wards on the low-intensity LLT prescription. In this subgroup, we found that 20% of patients with stroke/TIA discharged by the neurology ward received high-intensity therapy compared with 60% of patients with stroke/TIA discharged by cardiology ward. These findings suggest that the cardiologists, who presumably were more up-to-date on the efficacy of the LLT therapy, have a major awareness of ESC/EAS guideline recommendations.
We updated the discussion including the following sentence “Furthermore, analyzing the information about hospital wards (available for 54% of the study population) we found that the proportion of patients with stroke/TIA who received high-intensity statin therapy was 20% and 60% when they were discharged by the neurology and cardiology ward, respectively (data not shown). These findings may suggest that the cardiologists, who presumably were more up-to-date on the efficacy of the LLT therapy, may have a major awareness of ESC guidelines.”
Did the authors include in their analysis patients with all types of stroke? e.g lacunar, cardioembolic, large artery atherosclerosis
In our analysis we identified subjects with stroke/TIA if they have had an hospitalization for the following codes 433.xx (occlusion and stenosis of precerebral arteries), 434.xx (occlusion of cerebral arteries) and 435.xx (Transient cerebral ischemia) or a undergoing a procedure for the following codes 38.11 (Endarterectomy, intracranial vessels), 38.12 (Endarterectomy, other vessels of head and nec). Therefore, lacunar and cardioembolic stroke were included in the selection while the large artery atherosclerosis was not included. All the ICD-9 diagnosis and procedures codes used for the cohort selection are reported in the supplementary table 1.
Is there any information regarding LDL levels pre-hospitalization or post hospitalization? Is there any information regarding the percentage of patients that have achieved LDL targets?
Information about LDL levels was not available for our population throughout the entire study period. Therefore, the percentage of patients who achieved LDL targets is unknown. We updated the section of the study limit including the following sentence “As in all datasets of administrative data, type and number of clinical variables are limited. Particular disadvantage is the lack of LDL-c concentration throughout the entire study period.”
How many patients received PCSK9 therapy? All those patients were already under therapy with ezetimibe?
The number of patients treated with PCSK9 therapy was 64. Of these, 14 subjects were already under therapy with ezetimibe. We updated the results reported the absolute number for each LLT type.

Reviewer 2 Report
Reviewer
The article contains interesting material that fits very well with current research and the scope of secondary prevention of atherosclerotic cardiovascular disease (ASCVD). The aim of the study was to evaluate lipid-lowering therapy prescribing in patients discharge for an ASCVD event.
Title:
The title reflects well the scope of the research.
Abstract:
The abstract is clear and represents the scope of the study well.
Results:
I understand that the assumption of the study was not the assessment of LDL cholesterol level within 6 months from the discharge. Nevertheless, I am interested in the concentration of LDL cholesterol during hospitalization and the dose of cholesterol-lowering drugs recommended at hospital discharge. Were the LDL-c concentration during hospitalization and the dose of cholesterol-lowering drugs known at hospital discharge? Could these parameters affect the outpatient cholesterol lowering drugs dose?
Discussion:
Please refer to the current 2021 ESC Guidelines on cardiovascular disease prevention in clinical practice, European Heart Journal, Volume 42, Issue 34, 7 September 2021, Pages 3227–3337, https://doi.org/10.1093/eurheartj/ehab484, where the management is gradual and the goal is LDL cholesterol concentration.
Figure 2, 3.
The Figure legends should have a clear and descriptive title. Each legend should clearly describe what is being shown in the Figure. All abbreviations should be defined when first used in each Figure legend, so that the reader can understand the Figures when looking at them individually and separate from the main manuscript.
Author Response
I understand that the assumption of the study was not the assessment of LDL cholesterol level within 6 months from the discharge. Nevertheless, I am interested in the concentration of LDL cholesterol during hospitalization and the dose of cholesterol-lowering drugs recommended at hospital discharge. Were the LDL-c concentration during hospitalization and the dose of cholesterol-lowering drugs known at hospital discharge? Could these parameters affect the outpatient cholesterol lowering drugs dose?
Thanks for your comment. It would be interesting to explore how the LDL-c concentration during hospitalization and the cholesterol-lowering drugs dose recommended at hospital discharge affect the outpatient cholesterol lowering drugs dose, unfortunately the information was not available. We specified the lack of this information in the limit section including the following sentence: “As in all datasets of administrative data, type and number of clinical variables are limited. Particular disadvantage is the lack of LDL-c concentration throughout the entire study period. Further, the cholesterol-lowering drugs dose recommended at the hospital discharge was not available.”
Please refer to the current 2021 ESC Guidelines on cardiovascular disease prevention in clinical practice, European Heart Journal, Volume 42, Issue 34, 7 September 2021, Pages 3227–3337, https://doi.org/10.1093/eurheartj/ehab484, where the management is gradual and the goal is LDL cholesterol concentration.
As suggested, the discussion section was updated including the current 2021 ESC Guidelines.
The Figure legends should have a clear and descriptive title. Each legend should clearly describe what is being shown in the Figure. All abbreviations should be defined when first used in each Figure legend, so that the reader can understand the Figures when looking at them individually and separate from the main manuscript.
Thanks for your comments. As suggested, we updated the figure legends.
Reviewer 3 Report
The authors make their best effort to describe the regional data about high-intensity statin use in patients following discharge for atherosclerotic cardiovascular disease. Little new information and novelty were found. No detailed lipid -profile data before and after statin therapy shown in this study could be a big issue.
Author Response
The authors make their best effort to describe the regional data about high-intensity statin use in patients following discharge for atherosclerotic cardiovascular disease. Little new information and novelty were found. No detailed lipid -profile data before and after statin therapy shown in this study could be a big issue.
Thanks for your comment. With this study we aimed to provide updated data on the lipid management in an Italian ASCVD cohort also exploring the predictors of LLT intensity utilization. Specifically, we analysed the influence of the ASCVD conditions on the LLT intensity including coronary heart and cerebrovascular disease. To the best of our knowledge, evidence are limited after the 2019 ESC.
It would be interesting to explore the LDL-c concentration before and after the statin therapy, unfortunately the information about LDL value was not available for our cohort. We specified the lack of data in the limit section as follows: “As in all datasets of administrative data, type and number of clinical variables are limited. Particular disadvantage is the lack of LDL-c concentration throughout the entire study period.”

Round 2
Reviewer 3 Report
The limitation of no lipid profiles before and after LLT is a big issue and could not be accepted for publication in the current form.